# Caffeic Acid Phenethyl Ester Rescues Pulmonary Arterial Hypertension through the Inhibition of AKT/ERK-Dependent PDGF/HIF-1α In Vitro and In Vivo

**DOI:** 10.3390/ijms20061468

**Published:** 2019-03-22

**Authors:** Chin-Chang Cheng, Pei-Ling Chi, Min-Ci Shen, Chih-Wen Shu, Shue-Ren Wann, Chun-Peng Liu, Ching-Jiunn Tseng, Wei-Chun Huang

**Affiliations:** 1Department of Critical Care Medicine, Kaohsiung Veterans General Hospital, Kaohsiung 81362, Taiwan; cccheng@vghks.gov.tw (C.-C.C.); aqwert0216@gmail.com (M.-C.S.); cpliu@vghks.gov.tw (C.-P.L.); 2School of Medicine, National Yang-Ming University, Taipei 11221, Taiwan; 3Department of Physical Therapy, Fooyin University, Kaohsiung 83102, Taiwan; 4Department of Medical Education and Research, Kaohsiung Veterans General Hospital, Kaohsiung 81362, Taiwan; chi542738@gmail.com; 5Department of Pathology and Laboratory, Kaohsiung Veterans General Hospital, Kaohsiung 81362, Taiwan; 6Graduate Institute of Clinical Medicine, National Yang-Ming University, Taipei 11221, Taiwan; srwann01@mail.vhlc.gov.tw; 7School of Medicine for International Students, I-Shou University, Kaohsiung 82445, Taiwan; cwshu@isu.edu.tw; 8Kaohsiung Veterans General Hospital, Pingtung Branch, Pintung 91245, Taiwan; 9Institute of Biomedical Sciences, National Sun Yat-Sen University, Kaohsiung 80424, Taiwan; 10Department of Medical Research, China Medical University Hospital, China Medical University, Taichung 40402, Taiwan; 11School of Medicine, Kaohsiung Medical University, Kaohsiung 80708, Taiwan

**Keywords:** Caffeic acid phenethyl ester, hypoxia-inducible factor-1α, pulmonary arterial hypertension, pulmonary smooth muscle cell, apoptosis resistance

## Abstract

Pulmonary arterial hypertension (PAH) is characterized by pulmonary arterial proliferation and remodeling, resulting in a specific increase in right ventricle systolic pressure (RVSP) and, ultimately right ventricular failure. Recent studies have demonstrated that caffeic acid phenethyl ester (CAPE) exerts a protective role in NF-κB-mediated inflammatory diseases. However, the effect of CAPE on PAH remains to be elucidated. In this study, monocrotaline (MCT) was used to establish PAH in rats. Two weeks after the induction of PAH by MCT, CAPE was administrated by intraperitoneal injection once a day for two weeks. Pulmonary hemodynamic measurements and pulmonary artery morphological assessments were examined. Our results showed that administration of CAPE significantly suppressed MCT-induced vascular remodeling by decreasing the HIF-1α expression and PDGF-BB production, and improved in vivo RV systolic performance in rats. Furthermore, CAPE inhibits hypoxia- and PDGF-BB-induced HIF-1α expression by decreasing the activation of the AKT/ERK pathway, which results in the inhibition of human pulmonary artery smooth muscle cells (hPASMCs) proliferation and prevention of cells resistant to apoptosis. Overall, our data suggest that HIF-1α is regarded as an alternative target for CAPE in addition to NF-κB, and may represent a promising therapeutic agent for the treatment of PAH diseases.

## 1. Introduction

Pulmonary arterial hypertension (PAH) is a life-threatening disorder, characterized by vascular remodeling of small pulmonary arteries, which leads to sustained elevation of mean pulmonary artery pressure, right ventricular (RV) afterload and hypertrophy, and eventually RV failure [1,2,3]. Aberrant proliferation and resistance of pulmonary arterial smooth muscle cells (PASMCs) to apoptosis in the tunica media of the pulmonary artery are promoted by various growth factors, and these phenomena are the major cause of vascular remodeling in the development of PAH [4]. Recent studies have suggested that micro-environmental hypoxia is involved in the initiation and progression of PAH and remodeling of the pulmonary vessels [5]. Moreover, hypoxia and monocrotaline (MCT) lead to the development of a pro-inflammatory microenvironment that could contribute to hypoxic vasoconstriction and remodeling [6]. Although it has been recognized that hypoxia plays a vital role in the development of PAH, molecular and cellular mechanisms driving vascular remodeling remain poorly defined. 

Hypoxia-inducible factor-1 (HIF-1), a heterodimeric transcription factor containing a constitutive β subunit and a regulatory α subunit, is the most recognized pathway adopted by cells in a hypoxic microenvironment [7]. Consistent with its role in cellular oxygen homeostasis, in response to hypoxia, the HIF-1α subunit translocates into the nucleus and conjugates with HIF-1β to activate genes associated with cell growth, cell cycle events and glycolytic metabolism [8,9]. Hypoxia-induced smooth muscle cells proliferation is inhibited by HIF-1α knock-down [10], suggesting that HIF-1 may participate in pulmonary vascular remodeling. Similarly, the platelet-derived growth factor (PDGF) signaling pathway was demonstrated to be involved in the vascular remodeling [11]. Moreover, the previous study shows that HIF-1α is associated with phosphatidylinositol 3-kinase (PI3K) and nuclear factor κB (NF-κB) signaling in PASMCs [12]. Although considerable efforts have been made to understand the cellular mechanisms of HIF-1α-associated pulmonary vascular remodeling and PASMC proliferation and migration, the relationship between HIF-1α and PDGF-BB in the MCT-induced rat PAH model remains controversial. 

Caffeic acid phenethyl ester (CAPE), a major active component in propolis, is widely known for its anti-inflammatory effect due to its ability to suppress NF-κB activity [3,13]. For instance, CAPE could inhibit the mitogen-activated protein kinases (MAPKs) and NF-κB signaling to reduce inflammatory gene expression in activated human mast cells [3], and could regulate the expression of inflammation-associated matrix metalloproteinases (MMPs) and tissue inhibitors of metalloproteinases (TIMPs) in LPS-stimulated human monocytes [14]. Besides, a variety of studies demonstrate that CAPE promotes cell-cycle arrest and inhibits prostate cancer cell growth through mediating the expression of p53, p21 and Skp2 genes [15]. Furthermore, CAPE also induces cellular apoptosis in C6 glioma cells through activating p75 NTR, MAPK and neutral-sphingomyelinase signaling [16]. In non-cancerous primary cells, CAPE shows the potential to inhibit the PDGF-BB-stimulated proliferation and migration of human coronary smooth muscle cells [17], and also decreases the VEGF expression in human retinal pigment epithelial cells by inhibiting the PI3K and HIF-1α signaling [18]. Collectively, these findings indicate that CAPE plays an important role in ameliorating some diseases. However, the effects of CAPE on the treatment of PAH is still unknown. 

In the present study, we intend to establish whether the inhibition of HIF-1α signaling pathways activation and vascular remodeling by CAPE may indeed result in the inhibition of MCT-induced PAH in rats. We report here for the first time that CAPE ameliorates MCT-induced right ventricular systolic pressure (RVSP) and vascular remodeling by decreasing HIF-1α expression and PDGF-BB production. Furthermore, CAPE inhibits hypoxia- and PDGF-BB-induced HIF-1α expression by decreasing the activation of the AKT/ERK pathway, which results in the inhibition of hPASMCs proliferation and prevention of cells resistant to apoptosis.

## 2. Results

### 2.1. CAPE Attenuates Pulmonary Hypertension and Right Ventricular Hypertrophy in MCT-Treated Rats

The effectiveness of MCT on the induction of PAH symptoms was reflected by the elevated right ventricular systolic pressure (RVSP) (19.04 ± 0.56 vs. 40.74 ± 4.90, *p* < 0.01) and right ventricular hypertrophy (0.24 ± 0.02 vs. 0.42 ± 0.06, *p* < 0.01) indicated by a significantly increased RV/(LV+S) (Figure 1A,B). Conversely, there was no increase in systolic pressure of left ventricle (LVSP) or body weight in MCT-injured rats compared with control rats treated with PBS (Figure 1C,D), confirming the development of PAH in MCT-treated rats. CAPE has been shown to possess antioxidant and immunomodulatory properties [19]. To investigate whether CAPE was capable of reversing pulmonary hypertensive changes once MCT-PAH had already been establishe4, MCT-injured rats were given daily intraperitoneal injections of 5 or 10 mg/kg of CAPE, for 14 days beginning 2 weeks after MCT injection. Four weeks after the last series of injections, CAPE significantly reduced MCT-induced RVSP (30.20 ± 1.58 and 25.30 ± 2.41, *p* = 0.0008 and *p* < 0.01) and right ventricular hypertrophy (0.34 ± 0.03 and 0.3 ± 0.02, *p* = 0.02 and *p* = 0.003) in a dose-dependent manner (Figure 1A,B). Furthermore, CAPE administration did not affect LVSP and body weight in comparison with the PBS- or MCT-treated rats (Figure 1C,D).

### 2.2. Caffeic Acid Phenethyl Ester (CAPE) Prevents Pulmonary Vascular Remodeling in Monocrotaline (MCT)-Induced Pulmonary Arterial Hypertension (PAH) Rat Model 

Vascular proliferation and remodeling are the hallmarks of PAH pathogenesis [20]. To explore the in vivo effects of CAPE on PAH progression, vascular remodeling changes in the vessel wall thickness was measured. Elastic Van Gieson staining showed the morphometric changes within the aorta in MCT-injured rats (Figure 2A). The media thickness and the ratio of media thickness to lumen diameter were significantly increased in aorta of rats with MCT treatment (Figure 2B,C). Moreover, administration of CAPE effectively prevented lumen diameter and wall thickening of pulmonary arterioles in MCT-induced PAH rats.

### 2.3. CAPE Attenuates HIF-1α and PDGF-BB Expression in MCT-Treated Rats

To explore the possible mechanisms underlying the protective effects of CAPE against MCT-induced pulmonary vascular remodeling, we examined the HIF-1α protein levels and secreted PDGF-BB levels in lung tissues and serum, respectively, from rats treated with MCT. As shown in Figure 3A,B, MCT increased HIF-1α protein expression and serum concentrations of PDGF-BB. Administration of CAPE 5 or 10 mg/kg/day significantly inhibited MCT-induced HIF-1α and PDGF-BB production. Similar to the results observed with lung tissue lysates, the immunohistochemical study indicates that obvious densitometry of HIF-1α staining is detected on the hypertrophic media of the pulmonary artery in MCT-induced rats, which is dramatically absent in the lung specimens of the PBS and CAPE groups (Figure 3C). These results implied that a decrease in expression of HIF-1α and PDGF-BB by CAPE treatment may play a vital role in preventing vascular remodeling in MCT-injected rats. 

### 2.4. CAPE Attenuates Hypoxia- and PDGF-BB-Induced HIF-1α Expression via the AKT/ERK Pathway in hPASMCs

HIF-1 activation is a mediator of physiological and pathophysiological responses to hypoxic conditions [21]. Since CAPE attenuated wall thickening of pulmonary arterioles and reduced expression of HIF-1α and PDGF-BB in MCT-induced PAH rats, the underlying cellular mechanism involved in the expression of HIF-1α associated with vascular remodeling in PAH was further investigated on in vitro cultured hPASMCs. We found that HIF-1α protein and mRNA expression were enhanced after either hypoxic or PDGF-BB treatment in hPASMCs. In contrast, administration of CAPE significantly inhibited both hypoxia- and PDGF-BB-induced HIF-1α protein and mRNA expression (Figure 4A–D). Because AKT activation has been increasingly recognized as a regulator of vascular remodeling and controversy persists regarding the role of AKT in HIF-1α regulation, we explored a possible link between AKT phosphorylation and HIF-1α expression in hypoxia- and PDGF-BB-induced hPASMCs and then assessed the effects of CAPE on HIF-1α expression. Our results revealed that cells treated with hypoxia or PDGF-BB stimulated phosphorylation of ERK, AKT, and NF-κB, which were inhibited by inhibitors of ERK (PD98059) and PI3-kinase (LY294002), respectively. However, pretreatment with PD98059 had no significant effect on hypoxia- or PDGF-BB-induced AKT phosphorylation, indicating that hypoxia- and PDGF-BB-induced AKT and NF-κB phosphorylation is mediated through ERK activation in hPASMCs. We also found that either hypoxia or PDGF-BB induced HIF-1α expression, which was attenuated by pretreatment with PD98059 and LY294002 (Figure 4E,F), suggesting that hypoxia and PDGF-BB induced HIF-1α expression via a pathway involving AKT and ERK. Moreover, CAPE has been reported to be a potent and specific inhibitor of NF-κB [22]. Therefore, we investigated the role of CAPE in hypoxia or PDGF-BB-mediated responses. As shown in Figure 4G–H, pretreatment with CAPE attenuated hypoxia- and PDGF-BB-induced AKT, ERK, and NF-κB phosphorylation. These results suggested that administration of CAPE attenuates hypoxia- and PDGF-BB-induced HIF-1α expression and that it is mediated through AKT/ERK inactivation in hPASMCs.

### 2.5. CAPE Reduces Hypoxia- and PDGF-BB-Induced Proliferation via Inhibiting AKT/ERK Pathway in hPASMCs

Vascular remodeling is a complex process that varies over time and is dependent on abnormal smooth muscle cell activation. It has been reported that MCT-induced PAH via increasing the levels of hydrogen peroxide, even though the mechanism by which this occurs remains unclear [23]. Moreover, intracellular hydrogen peroxide generation was increased in rat PASMCs exposed to a hypoxic environment, leading to an increased risk of developing PAH [24]. Thus, we investigated whether a decreased oxygen level contributes to vascular remodeling associated with PAH via promoting smooth muscle cell proliferation and apoptosis. In vitro, hPASMCs were maintained in either hypoxic atmosphere (3% O_2_ level) or in the addition of recombinant PDGF-BB, with or without CAPE treatment. By 3-(4,5-Dimethylthiazol-2-yl)-2,5-diphenyltetrazolium bromide (MTT) assay, hPASMCs showed robust proliferation within 72 h under both hypoxia and PDGF-BB conditions, whereas in the presence of CAPE, the proliferation of hPASMCs was significantly retarded in a time-dependent manner (Figure 5A,B). Besides, by the examination of BrdU incorporation, CAPE treatment showed the ability to inhibit DNA synthesis of hPASMCs under hypoxia or PDGF-BB stimulation (Figure 5C,D). Furthermore, this growth-arrest effect of CAPE was supported by the fact that increasing percentage of hPASMCs expressed intensive senescence-associated β-galactosidase activity (Figure 5E,F). These results provide strong evidence that either hypoxia or PDGF-BB could induce simultaneous increases in proliferation of hPASMCs, while pretreatment with CAPE could suppress the effects.

### 2.6. CAPE Promotes the Apoptosis of hPASMCs

Regarding the gradual decrease of cell viability observed in MTT assays, whether hPASMCs turn out apoptosis in the appearance of CAPE was evaluated using the terminal deoxynucleotidyl transferase dUTP nick end labeling (TUNEL) assay. As shown in Figure 6A,B, either in hypoxia or in PDGF-BB condition, pretreatment of hPASMCs with CAPE had a significantly higher proportion of apoptotic cells than cells without CAPE. We also examined cleaved caspase-3, an activated form of caspase-3 that acts as cell death-related protease at the most distal stage of the apoptosis pathway. As shown in Figure 6C,D, CAPE at a dose of 10 μM significantly increased cleaved caspase-3 protein expression in either hypoxia- or PDGF-BB-treated hPASMCs. The results demonstrated that CAPE promotes hPASMC apoptosis in response to hypoxia or PDGF-BB stimulation. 

## 3. Discussion

Abnormal proliferation of PASMCs is the major cause of pulmonary vascular remodeling in PAH. Divergent stimuli mediate the activation of HIF-1α to turn on genes associated with cell survival, proliferation, migration, or metabolism might play a critical role in the pathology of PASMCs and development of PAH. CAPE, a polyphenolic natural product, has numerous biological activities, including anti-inflammatory and anti-proliferative effects [3,13]. Clinically, it has been shown to inhibit the growth of tumor cells [25] and to inhibit pulmonary fibrosis [26]. In this study, we suggest that CAPE can possibly be used as a therapeutic for PAH. Here, we proved for the first time that in hPASMCs, CAPE could inhibit hypoxia- and PDGF-BB-induced proliferation and resistance to apoptosis by decreasing the Akt/Erk1/2/HIF-1α pathway, which ameliorates vascular remodeling. Furthermore, in vivo studies indicate that CAPE can repress pulmonary vascular remodeling in MCT-induced PAH by down-regulating HIF-1α expression and PDGF-BB production. 

Pulmonary artery remodeling is considered a major feature of pulmonary hypertension, with marked proliferation of PASMCs and apoptosis resistance. MCT-induced PAH in rats results from the pulmonary accumulation of pyrrolic metabolites leading to endothelial injury and vasculature remodeling [27]. This is also confirmed by our observation that the rat models of MCT-induced PAH employed in this study represent specific changes in the pressures of the right ventricle and medial hypertrophy relative to the control group. 

Perros et al. [28] noticed that PDGF-B mRNA was highly expressed in microdissected pulmonary arteries from patients with PAH compared to healthy donors. However, there was no statistical difference in the expression of PDGF-A mRNA between the two groups. Immunohistochemical stain showed that PDGF-B was mainly expressed in smooth muscle cells from patients with PAH. PDGF-A to a lesser extent was detected in perivascular cells within plexiform lesions of PAH patients. Recently, a newly published study has discovered that plasma levels of PDGF-B increased in a subpopulation of patients with PAH [29]. In the monocrotaline rat model, a marked increase in PDGF-B protein was detected (by immunohistochemistry) in the medial layer of small pulmonary arteries, and PDGF receptor antagonism reverses disease in animal models [30,31], suggesting that PDGF-B is implicated in vascular smooth cell proliferation. Our results confirm the previous observation that serum PDGF-BB concentration significantly increased in monocrotaline-treated rats. Treatment of pulmonary arterial smooth cells (PASMCs) with PDGF-BB exerts a proliferative effect on cells. Further analysis revealed that the administration of CAPE effectively prevented lumen diameter and wall thickening of pulmonary arterioles in MCT-induced PAH rats and ameliorated PDGF-BB-induced PASMCs proliferation.

Other groups have shown that HIF-1α is upregulated in chronic hypoxia- and MCT-induced pulmonary hypertension [32]. Consistently, we demonstrated that HIF-1α level was significantly increased in MCT-challenged rats, whereas the reverse effect occurred in CAPE-treated rats. In this study, we report, for the first time, both hypoxia and PDGF-BB induced cell proliferation and HIF-1α protein expression in hPASMCs. On the other hand, HIF-1α and PDGF-BB induced resistance to apoptosis by interfering with caspase-3 activation in hPASMCs. The above description allows us to suggest that both HIF-1α and PDGF-BB are closely related to vascular remodeling in the development of PAH.

Recently, as CAPE, a potent and specific inhibitor of activation of NF-κB, has been proven to have antioxidant [33], and anti-inflammatory effects [34], more and more researchers have begun to study the cytoprotective mechanisms of CAPE. Song et al. indicated that CAPE and NAC exert a similar effect on suppression of H_2_O_2_-induced TNF-α mRNA expression in human middle ear epithelial cells [35]. Moreover, CAPE could inhibit IL-1β-induced IL-6, MCP-1 and ICAM-1 expression in human corneal fibroblasts [36]. However, there is little known about the effect of CAPE on vascular remodeling in PAH, and the exact mechanisms of action remain controversial. In this study, we showed that CAPE pretreatment markedly reduced RVSP and RV hypertrophy in MCT-treated rats. We also demonstrated that CAPE could decrease hypoxia- and PDGF-BB-induced cell proliferation in hPASMCs. Moreover, CAPE could significantly decrease MCT-enhanced the serum levels of PDGF-BB and HIF-1α expression in the lung tissues of rats. CAPE has been shown to mediate protection of ischemia/reperfusion injury through induction of HIF-1α [37]. Also, CAPE-induced HIF-1α expression effectively suppressed polyaromatic hydrocarbon-mediated hepatocarcinogenesis [38]. However, in this study, we proved that CAPE could perform its cytoprotective effect by inhibiting the expression of HIF-1α in response to hypoxia and PDGF-BB in hPASMCs and MCT-treated rats. The above results also allow us to prove that CAPE could be an effective approach in treating PAH via inhibition of the expression of HIF-1α and PDGF-BB. In fact, the HIF-1α/PDGF axis has been reported to be involved in virus-associated pulmonary vascular remodeling [39]. An interesting topic for future research is on the effects of CAPE on the correlation between HIF-1α and PDGF-BB in remodeled human pulmonary arteries. On the other hand, CAPE enhanced the activation of effector caspase-3 and -7, and potentiated TRAIL-induced apoptosis in hepatocellular carcinoma cells [40]. This was confirmed by our observation that hypoxia- and PDGF-BB-stimulated PASMCs showed resistance to apoptosis, whereas CAPE pretreatment promoted apoptosis by increasing the caspase-3 activity of hPASMCs.

HIF-1-modulated signaling plays an important role in the regulation of cell proliferation and anti-apoptosis [41]. In general, HIF-1α activation is regulated through various signaling pathways, such as NF-κB, AKT, Rac, and MAPKs [42,43,44]. In mouse PASMCs, hypoxia-induced HIF-1α mRNA expression via the PI3K/AKT pathway and activation of the NF-κB, but not the ERK pathway [12]. However, CoCl_2_-induced hypoxia promoted cell proliferation through the ERK and PI3K/AKT pathways [45]. On the other hand, HIF-1α was involved in hypoxia-induced rat PASMCs proliferation via AKT [46]. This is confirmed by our observation that hypoxia and PDGF-BB induced HIF-1α expression via the AKT/ERK pathway in hPASMCs. It is worth noting that CAPE attenuated phorbol myristate acetate and calcium ionophore A23187-stimulated phosphorylation of JNK, but not ERK and p38MAPK [3]. By contrast, CAPE inhibited the inflammatory effects of IL-1β by inhibiting the activation of AKT and NF-κB, whereas the activity of MAPKs was not affected [36]. In this study, we proved that CAPE could perform its cytoprotective effect by inhibiting the activation of AKT and ERK in response to hypoxia and PDGF-BB in hPASMCs. 

In summary, our results demonstrate that CAPE improves vascular remodeling in MCT-treated PAH models through down-regulation of HIF-1α and PDGF-BB. In hPASMCs, CAPE inhibits hypoxia- and PDGF-BB-induced AKT/ERK/HIF-1α signaling pathways and blocks excessive cell proliferation. CAPE prevents the resistance of hPASMCs to apoptosis through inhibition of caspase-3 activation. Our results so far provide molecular mechanisms for anti-vascular remodeling of CAPE, which seems to adjust between proliferation and apoptosis of vascular smooth muscle cells. Although we have provided a novel molecular mechanism underlying the anti-PAH effects of CAPE in vivo in rats, we believe much more research work is still needed to extrapolate these findings to humans. As Provencher [47] noted in his review of translational research in PAH, animal models are used sparingly in PAH studies due to both genetic and epigenetic differences in the animals and the incomplete reappearance of human phenotypes. Therefore, the effect of CAPE in different rat models of PAH should be undertaken to explore our hypothesis further.

## 4. Materials and Methods

### 4.1. Reagents and Antibodies

CAEP (#C8221), crotaline (#C2401), 4′,6-diamidino-2-phenylindole, dilactate (#D9564) and 5-bromo-2′-deoxyuridine (#B5002) were purchased from Sigma-Aldrich (Louis, MO, USA). Rabbit anti-Akt (#99272), anti-phospho-Akt (Ser473, #4060), anti-phospho-ERK1/2(Thr202/Tyr204, #4370) and anti-phospho-NF-κB (Ser536, #3033) were from Cell Signaling (Danvers, MA, USA), respectively. Mouse anti-BrdU (sc-32323) and rabbit anti-ERK2 (sc-154), anti-NF-κB p65 (sc-372) were from Santa Cruz Biotechnology (Santa Cruz, CA, USA), respectively. Rabbit anti-β-actin (ab8227) and anti-active/pro Caspase 3 (ab47131) were from Abcam (Cambridge, UK), respectively. Mouse monoclonal antibodies against the HIF-1α (NB100-105) was purchased from Novus Biological (Littleton, Arapahoe, CO, USA). Human recombinant PDGF-BB (SRP3138) was purchased from Sigma-Aldrich. PD98059 and LY 294002 were obtained from Biomol (Plymouth Meeting, PA, USA).

### 4.2. Cell Culture

Primary human pulmonary arterial smooth muscle cells (hPASMCs) purchased from Lonza Ltd. (Basel, Switzerland) was maintained in SmGM-2 medium including bullet kit supplements (Lonza, Visp, Switzerland). The cultured condition and procedure were described as previously published reference [48]. Experiments were performed using cells from passages 4–10. Hypoxia (3% O2) was achieved in the incubator by flowing nitrogen gas into it, and the oxygen concentration within was monitored continuously with an oxygen sensor.

### 4.3. MCT Rat Studies

PAH was incubated in male Sprague-Dawley rats (BioLASCO Co., Ltd, Taipei, Taiwan) weighing 200–250 g by a single subcutaneous injection of MCT (Sigma-Aldrich) at 60 mg/kg. Therapeutic intervention was performed at days 14 after MCT injection and treated with the daily intraperitoneal administration of 5 or 10 mg/kg CAPE. Animals were sacrificed at day 28, and PAH pathology was assessed as described previously [49]. All animal experiments were approved by the Institutional Animal Care and Use Committee, Kaohsiung Veterans General Hospital as well as confirmed to IACUC Guide.

### 4.4. Hemodynamic Measurements

Briefly, rats were incubated with a 16-gauge intravenous cannula for mechanical ventilation after anesthetized with Zoletil^®^ 50 (Virbac, Colombia, IN, USA). To evaluate pulmonary artery pressure (PAP), right ventricular systolic pressure (RVSP) and the left ventricular systolic pressure (LVSP) were measured by the polyethylene-50 tubing connected to a 24-gauge needle through the PowerLab/8_SP_ (AD Instruments Ltd. Dunedin, New Zealand). After hemodynamic measurement, animals were exsanguinated, and lung tissue was collected for histological and molecular profiling. The ratio of RV weight to body weight (RV/BW) and the ratio of RV weight to left ventricular (LV) plus septal weight (RV/LV+S) were calculated to provide Fulton index measurements [49,50].

### 4.5. Histology and Immunohistochemical Analysis of Pulmonary Arteries 

Lung specimens were washed with phosphate-buffered saline through the pulmonary artery, fixed in 4% formalin and embedded in paraffin. Tissue sections (4 μM) were stained with hematoxylin and eosin (H&E) or with Elastica van Gieson for morphometric analysis. For each artery, the percentage of medial wall thickness was measured at the two ends of the shortest external diameter of the distal pulmonary artery, and the average was taken as previously described [51]. Immunohistochemistry staining of paraffin-embedded lung sections was stained with anti-HIF-1α antibody followed by HRP-conjugated goat anti-mouse IgG2b (Santa Cruz). The immunostained samples mounted with Aquatex (Sigma-Aldrich) were examined under a LEICA microscopy (Leica, Wetzlar, Germany). Images were quantified with an image analyzer (LEICA Q550 IWB).

### 4.6. Cell Viability Assay

Cell viability was determined by MTT assay. Briefly, cells were plated into a 96-well plate at the concentration of 2 × 10^4^/mL. At different time point after hypoxia or PDGF-BB treatment in the presence or absence of CAPE, MTT reagent (5 mg/mL, 20 µL/well) was added to the culture medium and incubated with the cells for 4 h. Then 200 µL of 90% (*v*/*v*) DMSO was added to each well, and the plates were shaken for 15 min at room temperature. The number of viable cells was determined by optical density at 490 nm using a microplate reader (Thermo Fisher Scientific, Waltham,, MA, USA).

### 4.7. BrdU Proliferation Assay

Cells were plated on coverslips and treated as described above. Cells proliferation was determined using a BrdU labeling kit (Roche, Indianapolis, IN, USA). The BrdU incorporation was detected by FITC-BrdU antibody (Roche) and analyzed under a fluorescent microscope. The experiments were performed in triplicate.

### 4.8. Senescence-Associated β-Galactosidase (SA-β-gal) Staining

Senescence-associated β-galactosidase (SA-β-gal) activity was stained by Senescence Detection kit (Abcam) according to the manufacturer’s instruction. Briefly, hPASMCs were fixed with a fixative solution for 15 min, washed with PBS, and incubated with staining solution containing 1 mg/mL X-gal overnight. Images of random views were captured, and the blue cells were calculated.

### 4.9. Transferase-Mediated Deoxyuridine Triphosphate-Biotin Nick End Labeling (TUNEL) Assays

Immunofluorescence staining of apoptotic cells was performed by transferase-mediated deoxyuridine triphosphate-biotin nick end labeling (TUNEL) using the TUNEL Assay Kit (Roche, US). Briefly, hPASMCs were grown on poly-d-lysine pre-coated coverslips and incubated with PDGF-BB or under hypoxia (3% O_2_) condition, followed by fixation with 4% paraformaldehyde in PBS. Cells were washed with PBS, transferred to the equilibration buffer supplied in the kit, and incubated with the Label solution in a humidified chamber at 37 °C for 1 h. After incubation, cells were washed with PBS and mounted with Aquatex reagent with DAPI. TUNEL-positive hPASMCs (%) were quantified by automated counting performed by image analysis software (ImageJ, NIH). The apoptosis rate was counted from three independent experiments, and five microscopic fields were calculated in each slide.

### 4.10. Quantitative PCR (qPCR)

Total RNA was isolated from cell cultures using Trizol (Life Technologies, Waltham, MA, USA) and quantified by a NanoDrop spectrophotometer (Thermo Scientific, Waltham, MA, USA). Isolated RNA was reverse-transcribed using High-Capacity cDNA Reverse Transcription kit (Applied Biosystems, Foster, CA, USA) according to the manufacturer’s instructions. qPCR analyses were performed with Fast SYBR^®^ Green Master Mix (Applied Biosystems, Foster, CA USA) on a StepOnePlus^TM^ System (Applied Biosystems). All data were normalized to β-actin levels and were analyzed using the comparative Ct method. For qPCR, the following primer sequences were used: human 36B4 forward, 5′-GCCAGCGAAGCCACGCTGCTGAAC-3′ and reverse, 5′-CGAACACCTGCTGGATGACCAGCCC-3′; and HIF-1α forward, 5′-ACAAGCCACCTGAGGAGAGG-3′ and reverse, 5′-GAAGAGAAGGAAAGGCAAGTCC-3′.

### 4.11. Western Blot

Human PASMCs were lysed in RIPA lysis buffer with a protease inhibitor cocktail (Sigma Aldrich), and protein content was quantified by using the Pierce BCA Protein Assay Kit (Thermo Fisher Scientific, Waltham, MA, USA). 20 μg of protein was subjected to SDS-PAGE analysis. PVDF membranes were incubated with primary antibodies and incubated with HRP-conjugated secondary antibodies, followed by detection with ECL reagents (Bio-Rad Laboratories, Hercules, CA, USA). 

### 4.12. Measurement of PDGF-BB Generation

Rats were treated with PBS or CAPE (5 or 10 mg/kg) from days 14–28 after MCT treatment 14 days (60 mg/kg). After the end of treatment, serum PDGF-BB level was measured with an ELISA kit (R&D Systems, Minneapolis, MN, USA). Optical density was determined on a plate reader at an absorbance of 450 nm with wavelength correction at 540 nm to correct for the optical imperfections in the plate.

### 4.13. Statistical Analysis

Three independent performances were carried out for each experiment. For animal study, ten rats in each group were subjected to investigation. Data were expressed as mean ± SEM. One-way ANOVA followed by post hoc test of the Scheffé method was used for multiple comparisons. Significance was defined as *p* < 0.05. All statistical analyses were performed by the IBM^®^ SPSS^®^ Statistics Version 21 software.

## Figures and Tables

**Figure 1 ijms-20-01468-f001:**
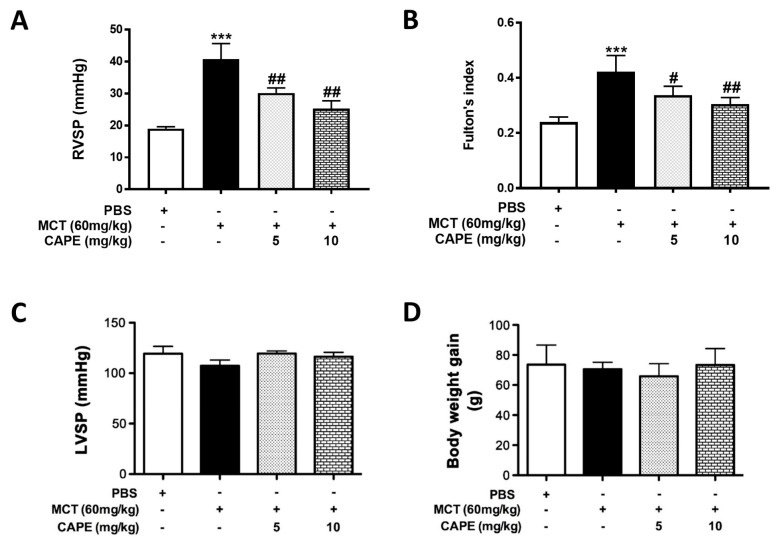
Caffeic acid phenethyl ester (CAPE) improves monocrotaline (MCT)-induced pulmonary arterial hypertension (PAH) in rats. (**A**) Rats were treated with CAPE (*n* = 6, 5 or 10 mg/kg) from day 14 to 28 after MCT injection (60 mg/kg). The rats in the healthy group received PBS injection instead of MCT (*n* = 5). Assessment of right ventricular systolic pressure (RVSP), (**B**) right ventricular hypertrophy (Fulton index, the ratio of right ventricular weight to left ventricular plus septal weight, (**C**) left ventricular systolic pressure (LVSP), and (**D**) body weight in rats. Data in A and B are expressed as mean ± SEM of five independent experiments. *** *p* < 0.01, as compared with the PBS group. ^#^
*p* < 0.05; ^##^
*p* < 0.01, as compared with the rats exposed to MCT alone.

**Figure 2 ijms-20-01468-f002:**
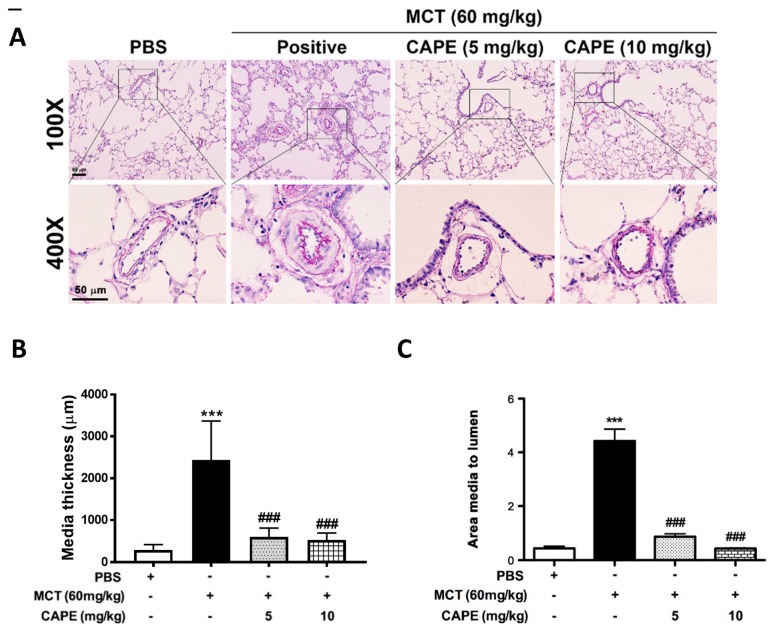
CAPE treatment reverses vascular remodeling in MCT-induced PAH rats. (**A**) Representative images of media hypertrophy in lung sections from the rats described in (**A**). Lung sections were stained for Elastic van Gieson (EvG). High magnification of images derived from the blocks is further displayed. Scale bars, 50 μm. (**B**,**C**) The degree of vascular remodeling was evaluated by the pulmonary arterial wall thickness and the ratio of media thickness to lumen diameter using Image J analysis software. Values are means ± SEM. *** *p* < 0.01 vs. PBS group; ### *p* < 0.01 vs. MCT group. *N* = 10 independent images of arteries per group from five animals.

**Figure 3 ijms-20-01468-f003:**
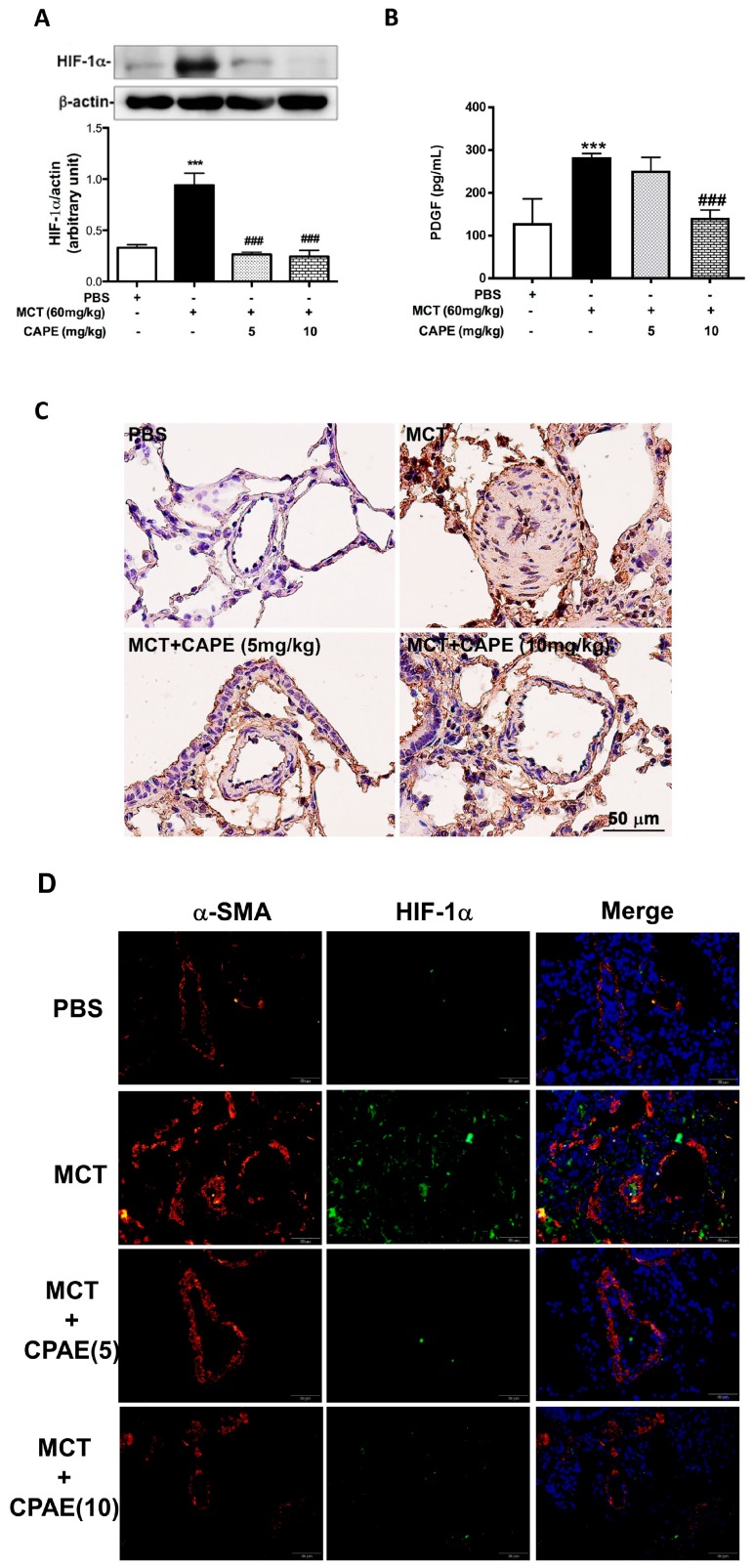
CAPE inhibits MCT-induced HIF-1α expression and PDGF-BB production in rats. (**A**) Rats were treated with PBS or CAPE (5 or 10 mg/kg) from days 14–28 after MCT treatment 14 days (60 mg/kg). Preparation of lung tissues was analyzed by Western blot to determine the levels of HIF-1α protein. Glyceraldehyde-3-phosphate dehydrogenase (GAPDH)was used as a loading control. (**B**) Serum levels of PDGF-BB in rat was assayed by an ELISA kit. (**C**) Representative images of immunohistochemical staining for HIF-1α in lung sections from the rats. Scale bars, 50 μm. (**D**) Immunofluorescence staining of alpha-smooth muscle actin (α-SMA) and HIF-1α in lung sections from rats. Scale bars, 50 μm. Data in (**A**,**B**) are expressed as mean ± SEM of three independent experiments. *** *p* < 0.01, as compared with the PBS group. ### *p* < 0.01, as compared with the rats exposed to MCT alone. Results are representative of three rats per experimental group.

**Figure 4 ijms-20-01468-f004:**
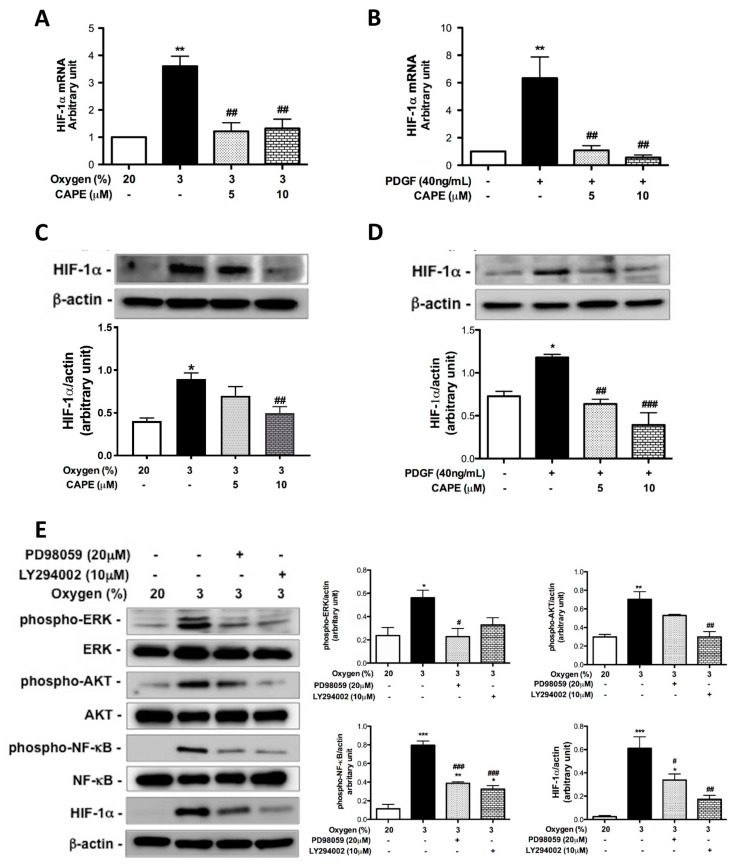
CAPE inhibits hypoxia- and PDGF-BB-induced HIF-1α expression through a AKT/ERK pathway in human PASMCs. (**A**) Cells were cultured for 24 h in normoxia (20%) or hypoxia (3%) with or without CAPE (5 or 10 μM). (**B**) Cells were pretreated with CAPE for 2 h, and then treated with or without PDGF-BB for 24 h. The HIF-1α mRNA levels were determined by real-time PCR. (**C**) Cells were cultured for 6 h in normoxia (20%) or hypoxia (3%) with or without CAPE (5 or 10 μM). (**D**) Cells were pretreated with CAPE for 2 h, and then treated with or without PDGF-BB for 6 h. The HIF-1α expression was determined by Western blot. (**E**–**H**) Production of phospho-ERK, AKT, NF-κB and expression of HIF-1α by human PASMCs. Cells were pretreated in the absence of inhibitor or with PD98059 (20 μM), LY294002 (10 μM), or CAPE (5 or 10 μM) for 2h, and then were cultured for 4 h in normoxia (20%), hypoxia (3%), or PDGF-BB. Cell lysates were examined by Western blotting using specific antibodies. Values are the mean ± SEM of three independent experiments. * *p* < 0.05; ** *p* < 0.002; *** *p* < 0.001, as compared with the control group. ^#^
*p* < 0.05; ^##^
*p* < 0.002; ^###^
*p* < 0.001, as compared with the cells exposed to hypoxia or PDGF-BB.

**Figure 5 ijms-20-01468-f005:**
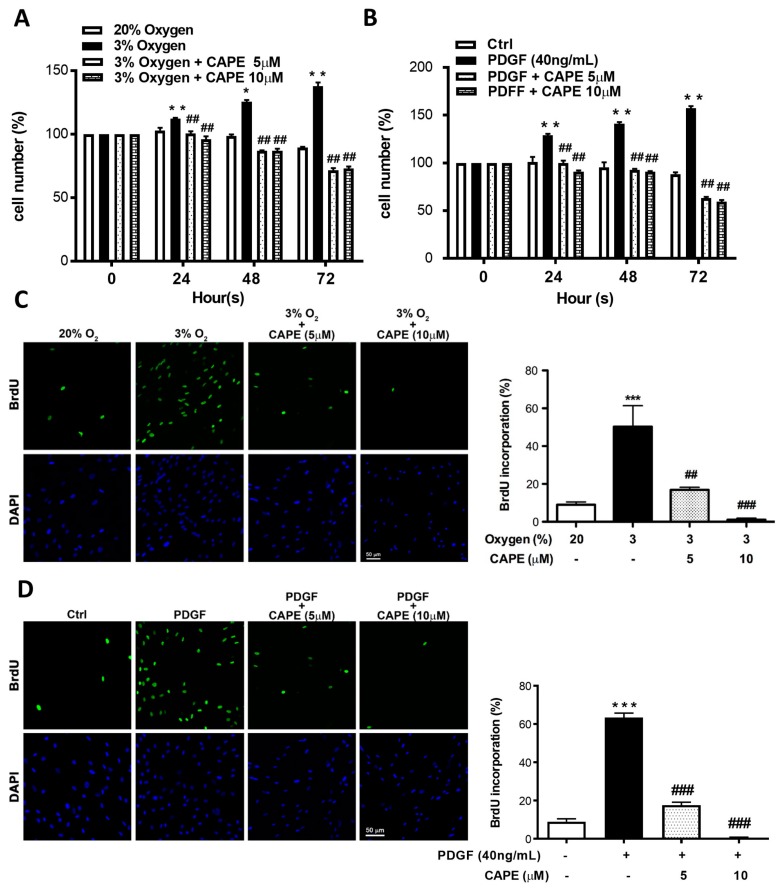
CAPE inhibits hypoxia- and PDGF-BB-induced cell proliferation and increases SA-β-gal activity of human PASMCs. (**A**,**B**) Cells were pretreated in the absence of inhibitor or with CAPE (5 or 10 μM) for 2 h, and then were cultured for the indicated times in normoxia (20%), hypoxia (3%), or PDGF-BB. Cell viability was determined by 3-(4,5-Dimethylthiazol-2-yl)-2,5-diphenyltetrazolium bromide (MTT) assay. (**C**,**D**) Cells were pretreated in the absence of inhibitor or with CAPE for 2 h, and then were cultured for 24 h in normoxia (20%), hypoxia (3%), or PDGF-BB. The proliferation of human PASMCs stained with BrdU/DAPI. All cell nuclei were stained with DAPI in blue, and the dividing cells were immunostained with anti-BrdU antibody in green. The percentage of BrdU-positive cells was presented. Scale bars, 50 μm. (**E**,**F**) SA-β-Gal staining of human PASMCs cultured for 24 h in normoxia (20%) or hypoxia (3%) with or without CAPE (5 or 10 μM). Photography images and quantification of SA-β-Gal positive cells are shown. Values are the mean ± SEM of three independent experiments. * *p* < 0.05; ** *p* < 0.002; *** *p* < 0.001, as compared with the control group. ^#^
*p* < 0.05; ^##^
*p* < 0.002; ^###^
*p* < 0.001, as compared with the cells exposed to hypoxia or PDGF-BB.

**Figure 6 ijms-20-01468-f006:**
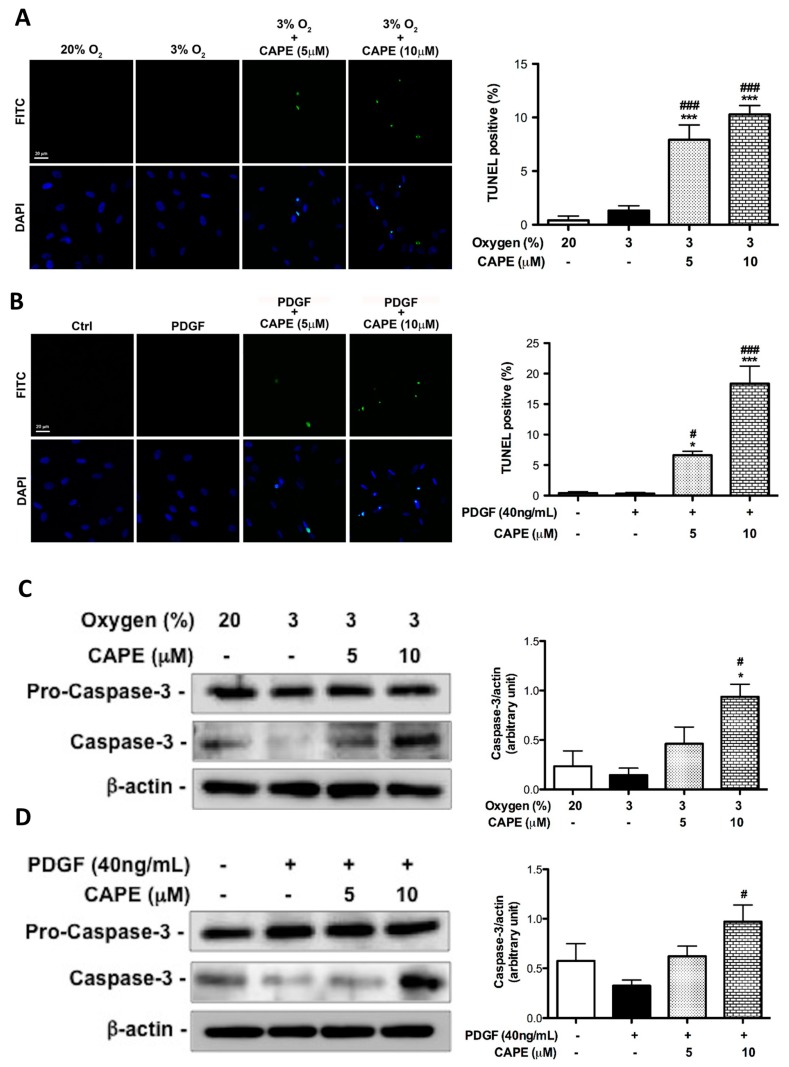
Up-regulation of caspase3 by CAPE promotes hypoxia- and PDGF-BB-induced resistance to apoptosis human PASMCs. (**A**,**B**) Cells were pretreated in the absence of inhibitor or with CAPE (5 or 10 μM) for 2 h, and then were cultured for 24 h in normoxia (20%), hypoxia (3%), or PDGF-BB. Cell apoptosis as measured by TUNEL assay. Apoptosis rate was calculated as a percent of TUNEL-positive cells out of a total number of cells indicated by DAPI-positive staining for each microscopic field. (**C**) Cells were cultured for 6 h in normoxia (20%) or hypoxia (3%) with or without CAPE (5 or 10 μM). (**D**) Cells were pretreated with CAPE for 2 h, and then treated with or without PDGF-BB for 6 h. Procaspase-3 and activated caspase-3 protein expression was evaluated by Western blot. Values are the mean ± SEM of three independent experiments. * *p* < 0.05; *** *p* < 0.01, as compared with the control group. ^#^
*p* < 0.05; ^###^
*p* < 0.01, as compared with the cells exposed to hypoxia or PDGF-BB.

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
