# Peer review of "Caffeic Acid Phenethyl Ester Rescues Pulmonary Arterial Hypertension through the Inhibition of AKT/ERK-Dependent PDGF/HIF-1α In Vitro and In Vivo"

_ijms, 2019, doi:10.3390/ijms20061468_

Round 1

Reviewer 1 Report

In this manuscript, authors used rats, treated with monocrotaline (MCT), to create a pulmonary arterial hypertension (PHA) model system. They demonstrated that administration of caffeic acid phenethyl ester (CAPE; 5 or 10 mg/kg) can attenuate the MCT-induced PHA, hypertrophy of the right heart ventricle, as well as the aorta and pulmonary vascular remodeling. They further showed that CAPE diminished the increases of HIF-1a and PDGF-BB expression in MCT treatment rats.

Then the authors utilized primary cultures of human pulmonary arterial smooth muscle cells with 3% oxygen (hypoxia) condition. They showed that the increase of HIF-1a and PDGF-BB expression occurred under 3% oxygen, which can be attenuated by treated with CAPE 5 or 10mM. The treatment of CAPE can block ERK, AKT and NF-kB phosphorylations induced by 3% oxygen and PDGF (40ng/ml) treatment, respectively. CAPE treatment reduced the cell proliferation and cell numbers, but increased caspase 3 expression and apoptosis.

This is an interesting study. However, additional experiments/data should be provided in order to fully support their conclusion and to provide a direct connection between results obtained from the MCT murine model and human pulmonary arterial smooth muscle cells.

Main concerns:

1)     In Fig. 3, using rat lung tissues, the authors showed an increase of HIF-1a level. There are many different types of cells present in lung tissues; therefore, there is no direct evidence to demonstrate the increase of HIF-1a level happening in pulmonary arterial smooth muscle cells. The authors should provide the direct evidence, such as using Western blot with pulmonary arterial smooth muscle cell lysates, using co-immunostaining technique, etc.

2)     Similarly, the increase of PDGF-BB level was detected by ELISA using rat serum (Fig 3). Is there any change of PDGF receptor level in both rat and human pulmonary arterial smooth muscle cells under MCT and hypoxia (3% oxygen) condition? How are the outcomes with or without CAPE treatment?

Minor concerns:

1)     There are PDGF-AA, -BB, or –AB forms. The authors only examined PDGF-BB in this study. This discrepancy should narrate, explain and discuss.

2)     The authors should carefully check their figures and figure legends. For example, there are wrong labels (+, - signs) in Fig. 1A.

3)     There are different font sizes (such as in p.14) and repeated words (such as the word of “p21” in p.4).  The authors should also carefully check the entire manuscript.

Author Response

Main concerns:

1) In Fig. 3, using rat lung tissues, the authors showed an increase of HIF-1a level. There are many different types of cells present in lung tissues; therefore, there is no direct evidence to demonstrate the increase of HIF-1a level happening in pulmonary arterial smooth muscle cells. The authors should provide the direct evidence, such as using Western blot with pulmonary arterial smooth muscle cell lysates, using co-immunostaining technique, etc.

Response:

According to the Reviewer’s suggestion, we have performed and added some data concerning the expression pattern of HIF-1a in the lung tissue of rats with PAH. Immunofluorescence staining confirmed the attenuation of MCT-induced HIF-1a expression by CAPE. In addition, partial co-expression of a-smooth muscle actin (a marker of vascular smooth muscle cell) and HIF-1a was detectable in lung sections of MCT-challenged rats (Fig. 3 D).

See Fig. 3 D and Page 5, Line 24-27.

The levels of HIF-1a, a-smooth muscle actin (a-SMA), vascular endothelial cadherin (VE-cadherin) and vimentin in lung tissue homogenates were also measured by Western blotting (as follows). The results demonstrated that the vascular smooth muscle cell marker a-SMA and the endothelial cell marker VE-cadherin are present in lung tissue homogenate, whereas the fibroblast marker vimentin is less detectable. The observation that HIF-1a play dual roles in vascular remodeling, both by modulating vascular smooth muscle cell and endothelial cell. In the present study, we focused on the effect of CAPE on pulmonary smooth muscle cells in PAH. The effect of CAPE on modulating vascular endothelial cells could be an interesting issue in the future. Currently, we have tried our best to perform experiments with isolation of pulmonary vascular smooth muscle cells form rats with PAH to confirm these results.

2) Similarly, the increase of PDGF-BB level was detected by ELISA using rat serum (Fig 3). Is there any change of PDGF receptor level in both rat and human pulmonary arterial smooth muscle cells under MCT and hypoxia (3% oxygen) condition? How are the outcomes with or without CAPE treatment?

Response:

Several studies have shown increased expression of PDGF receptor b (PDGFR b) in patients with idiopathic PAH and also in MCT models of PAH (Mathew, 2012; Perros et al., 2008; Schermuly et al., 2005). Our results confirm the previous observation that MCT highly upregulated PDGF b protein expression in rat lung tissue. However, pretreatment with CAPE in MCT-challenged rats suppressed PDGFR b expression (as follows: Figure A). Also, the mRNA expression of PDGFR b were increased in PASMCs derived from MCT-treated rats (as follows: Figure B). To determine the effect of CAPE on PDGFR b expression in human PASMCs, cells were pretreated with or without CAPE and exposed to normoxia or hypoxia for 24 hr. The results showed that 10 mM CAPE could attenuate hypoxic increases in PDGFR b expression in human PASMCs (as follows: Figure C). As data of the effect of hypoxia on PDGFR b expression in human PASMCs remains elusive, cells will expose to hypoxia for various times in future to clarify the effect of hypoxia on PDGFR b expression and to further confirm the reverse therapeutic effect of CAPE.

Minor concerns:

1)      There are PDGF-AA, -BB, or -AB forms. The authors only examined PDGF-BB in this study. This discrepancy should narrate, explain and discuss.

Response:

Perros et al. (2008) noticed that PDGF-B mRNA was highly expressed in microdissected pulmonary arteries from patients with PAH compared to healthy donor. However, there was no statistical difference in the expression of PDGF-A mRNA between the two groups. Immunohistochemical stain showed that PDGF-B was mainly expressed in smooth muscle cells from patients with PAH. PDGF-A to a lesser extent was detected in perivascular cells within plexiform lesions of PAH patients. Recently, a new study published has discovered that plasma levels of PDGF-B increased in a subpopulation of patients with PAH (Sweatt et al., 2019). Moreover, in the monocrotaline rat model, a marked increase in PDGF-B protein was detected (by immunohistochemistry) in the medial layer of small pulmonary arteries, and PDGF receptor antagonism reverses disease in animal models (Sanchez et al., 2007; Schermuly et al., 2005), suggesting that PDGF-BB is implicated in vascular smooth cell proliferation. Our results confirm the previous observation that serum PDGF-BB concentration significantly increased in the monocrotaline-treated rats. Treatment of pulmonary arterial smooth cells (PASMCs) with PDGF-BB exerts a proliferative effect on cells. Further analysis revealed that administration of CAPE effectively prevented lumen diameter and wall thickening of pulmonary arterioles in MCT-induced PAH rats and ameliorated PDGF-BB-induced PASMCs proliferation.

We have added the information in “Discussion” according to the Reviewer’s suggestion.

See Page 8 and Line 23.

2) The authors should carefully check their figures and figure legends. For example, there are wrong labels (+, - signs) in Fig. 1A.

Response:

According to the Reviewer’s suggestion, we have corrected the error in Fig. 1A.

See Fig. 1A.

3) There are different font sizes (such as in p.14) and repeated words (such as the word of “p21” in p.4).  The authors should also carefully check the entire manuscript.

Response:

According to the Reviewer’s suggestion, we have corrected this mistake and carefully checked throughout the text.

See Page 17 and Line 1; Page 3 and Line 26-28.

Reviewer 2 Report

The manuscript by Cheng et al., demonstrates the application of CAPE in successfully suppressing the vascular remodeling induced by MCT and improving the condition of PAH by using in vivo rat model and primary human pulmonary arterial SMC. Also, the authors went on to explore the mechanisms behind the therapeutic aspect of CAPE and found that it prevents hypoxia and PDGF induced stimulation of HIF1-α through the inactivation of AKT/ERK signaling.  The study is well controlled and novel that identifies HIF1-α as an additional target and supports CAPE for a therapeutic opportunity against PAH. However, the following minor changes may be required.

Minor comments

1.       In the Fig 1A, the labelling for PBS (+ symbol) appears to be wrongly mentioned.

2.       Page 2, 3rd paragraph (7th line) appears to mention p21 twice.  Also in Page 2, the first line of the last paragraph can be re-written to make it clear.

3.       In the Fig 4H (Page no. 8, right side at the bottom), statistical significance for ## is mentioned in the figure but its corresponding description is missing in the figure legends.

4.       Page 10, Label for “E” is missing in the figure.

5.       In the figure 5 legends, the description for the labelling of “##” is missing.  

6.       Page 14, fonts in 4.1 section needed to me made uniform.

7.       Page 15, the percentage of DMSO used can be mentioned.

8.       The statistical analysis part under methods section mention that the experiments are +/- SD however the figure legends mention them to be +/- SEM. This needs to be clarified or corrected.

Author Response

Minor comments:

1.  In Fig. 1A, the labeling for PBS (+ symbol) appears to be wrongly mentioned.

Response:

According to the Reviewer’s suggestion, we have corrected the error in Fig. 1A.

See Fig. 1A.

2.  Page 2, 3rd paragraph (7th line) appears to mention p21 twice. Also in Page 2, the first line of the last paragraph can be re-written to make it clear.

Response:

According to the Reviewer’s suggestion, we have corrected this mistake and revised some sentences that have not been constructed well.

See Page 3, Line 26-28.

Caffeic acid phenethyl ester (CAPE), a biologically active ingredient of propolis from honeybee, which is mostly well-known for its anti-inflammatory effect via suppressing the activation of NF-kB [3, 13].

Caffeic acid phenethyl ester (CAPE), a major active component in propolis, is widely known for its anti-inflammatory effect due to its ability to suppress the NF-kB activity [3, 13].

See Page 3, Line 21-23.

3. In the Fig 4H (Page no. 8, right side at the bottom), statistical significance for ## is mentioned in the figure but its corresponding description is missing in the figure legends.

Response:

According to the Reviewer’s suggestion, we have added the information in the figure 4 legends.

See Page 24, Line 13-15.

4. Page 10, Label for “E” is missing in the figure.

Response:

It is a typo error. We have corrected this mistake in Section 2.6 “CAPE promotes the apoptosis of hPASMCs.

See Page 7, Line 29.

5. In the figure 5 legends, the description for the labelling of “##” is missing. 

Response:

According to the Reviewer’s suggestion, we have added the information in the figure 5 legends.

See Page 24, Line 30.

6. Page 14, fonts in 4.1 section needed to me made uniform.

Response:

According to the Reviewer’s suggestion and for not making readers confused, we have modified these sentences in Section 4.1.

See Page 11, Section 4.1. “Reagents and antibodies”.

7. Page 15, the percentage of DMSO used can be mentioned.

Response:

According to Reviewer’s suggestion, we have added information about the percent of DMSO in Section 4.6 “Cell viability assay”.

See Page 13, Line 10.

8. The statistical analysis part under methods section mention that the experiments are +/- SD however the figure legends mention them to be +/- SEM. This needs to be clarified or corrected.

Response:

It is a typo error. We have corrected this mistake in “Statistical analysis”.

See Page 15, Line 12.
